# Iron, Neuroinflammation and Neurodegeneration

**DOI:** 10.3390/ijms23137267

**Published:** 2022-06-30

**Authors:** Roberta J. Ward, David T. Dexter, Robert R. Crichton

**Affiliations:** 1Department of Medicine, Imperial College, London SW7 2AZ, UK; ddexter@parkinsons.org.uk; 2Parkinson’s UK, London SW1V 1EJ, UK; 3School of Chemistry, University Catholique de Louvain, 1348 Louvain-la-Neuve, Belgium; robert.crichton@uclouvain.be

**Keywords:** neuroinflammation, iron homeostasis, microglia, astrocytes, iron

## Abstract

Disturbance of the brain homeostasis, either directly via the formation of abnormal proteins or cerebral hypo-perfusion, or indirectly via peripheral inflammation, will activate microglia to synthesise a variety of pro-inflammatory agents which may lead to inflammation and cell death. The pro-inflammatory cytokines will induce changes in the iron proteins responsible for maintaining iron homeostasis, such that increased amounts of iron will be deposited in cells in the brain. The generation of reactive oxygen and nitrogen species, which is directly involved in the inflammatory process, can significantly affect iron metabolism via their interaction with iron-regulatory proteins (IRPs). This underlies the importance of ensuring that iron is maintained in a form that can be kept under control; hence, the elegant mechanisms which have become increasingly well understood for regulating iron homeostasis. Therapeutic approaches to minimise the toxicity of iron include N-acetyl cysteine, non-steroidal anti-inflammatory compounds and iron chelation.

## 1. Iron Metabolism and Homeostasis

In this introductory section, we will begin with a brief presentation of iron metabolism and homeostasis, firstly at a cellular, and then at a systemic level in general terms, based on our current understanding, before turning to the more difficult issue of iron metabolism and its regulation within the brain.

### 1.1. Iron Metabolism

Iron is transported in the systemic circulation by transferrin (Tf), a bilobal protein present at a concentration of 35 mM in human serum, with a binding site for one Fe^3+^ together with a carbonate anion in each lobe. Typically, serum transferrin is around 30 per cent saturated representing only 0.1 per cent of total body iron (4 mg), and is the component of total body iron metabolism with by far the greatest turnover (some 25 mg of iron per day). Most human cells acquire their iron from circulating diferric Tf (Fe_2_Tf), via the transferrin to cell cycle (Figure 1). Enteroctyes in the duodenum reduce dietary ferric iron to ferrous iron using Dcytb1 and import Fe(II) into the cells via DMT1. Non-transferrin-bound iron (NTBI) is normally only found in the circulation in conditions of iron overload, when Tf is almost saturated with iron. NTBI may be imported into cells via the Zinc–Iron-Proteins Zip8 and Zip14 or by DMT, (Fe(III) present in NTBI must first undergo reduction to Fe(II) by ferrireductases at the plasma membrane). Iron is incorporated into many proteins as a cofactor in the form of haem and FeS clusters (in haemoproteins and FeS proteins), or mono- and dinuclear iron centres incorporated into both mononuclear and dinuclear non-haem iron proteins. In the cytosol, iron is stored in a soluble, non-toxic, yet bio-available form in the iron storage protein, ferritin. Mammalian ferritins are hollow protein shells, made up of 24 subunits, within which a hydrated ferric oxide core is deposited, and are typically heteropolymers made up of two subunits with distinct amino acid sequences, designated H and L. The H chains are characterised by a di-iron ferroxidase centre involved in the oxidation of Fe(II) to Fe(III), whereas L chains are thought to be responsible for nucleation of the mineral iron core [1]. NCOA4 (Nuclear receptor coactivator 4), a selective cargo receptor, mediates the autophagic degradation of ferritin (“ferritinophagy”), by directing ferritin to the lysosome, where it is believed to form hemosiderin, the product of ferritin aggregation and degradation, found in lysosomes in conditions of iron overload. Ferritinophagy has been recently shown to regulate ferroptosis, a newly described form of iron-dependent cell death mediated by excess lipid peroxidation [2]. The only known system for iron export from cells is ferroportin (Fpn1), a multidomain transmembrane protein which is expressed on cells including enterocytes, hepatocytes and reticulo-endothelial macrophages. Once exported bound to Fpn1, iron is oxidised to ferric iron via the multicopper oxidases ceruloplasmin or hephaestin and loaded onto ApoTf to be delivered through the circulation to other cells. Within the cytosol, a kinetically Labile Iron Pool (LIP), coordinated by small molecular weight chelating agents and iron chaperones (PCBPs), is present at concentrations ranging from 1 to 7 µM. PCBPs (Poly rC binding proteins) are multifunctional proteins that can coordinate single-stranded DNA and RNA as well as binding iron complexes [3]. Pcbp1 delivers iron to ferritin [4] and to the non-haem iron enzymes in the cytosol, whereas Pcbp2 interacts with membrane proteins to facilitate Fe(II) trafficking in and out of the cytosol, binding to iron-loaded DMTI and to Fpn1 [5]. In haemolytic conditions, haemoglobin and haem released in circulation are, respectively, bound to haptoglobin (HP) and haemopexin (HPX) to be internalised by the cell. HO-1 cleaves haem to form biliverdin and releases ferrous iron. Newly synthesised haem is exported via FLVCR [6].

### 1.2. Intracellular Iron Homeostasis

Intracellular iron homeostasis is controlled by the IRE/IRP system (Figure 1), consisting of two RNA-binding proteins (Iron Regulatory Proteins (IRPs)), IRP1 and IRP2, which bind specifically to Iron Regulatory Elements (IREs) located within target messenger RNA (mRNA) molecules [7]. In conditions of iron deficiency, IRP1 andIRP2 are activated for binding to the IREs in the untranslated regions (UTRs) of the mRNAs encoding for ferritin, Fpn1 and TfR1. Ferritin (both H- and L-chain) mRNAs contain a single IRE in their 5′ UTR, as does Fpn1, and binding of the IRPs to their IREs blocks their synthesis by preventing the recruitment of the small ribosomal subunit to the mRNA thereby blocking initiation of their translation [8]. TfR1 mRNA has multiple IREs in its 3′ UTR and the binding of IRPs protects the mRNA against nucleolytic degradation. These responses promote uptake of iron and prevent its storage or efflux. In contrast, in iron-replete cells the lack of IRE/IRP interaction allows the synthesis of ferritin and Fpn1 to take place, and results in the degradation of TfR1 mRNA, which appears to require the endonuclease regnase-1 [9]. As a result, iron uptake is prevented, and excess iron is stored or exported. A number of other IRE-containing mRNAs have been identified [7]. A single 3′ UTR IRE has been found in the mRNA encoding the metal transporter DMT1, which appears to contribute to its post-transcriptional regulation. A single ‘ferritin-like’ 5′UTR IRE was found in the mRNAs for hypoxia inducible factor *2α*(HIF-2α), involved in regulation of cellular responses to hypoxia, 5-aminolevulinic acid synthase 2(ALAS2), the first enzyme of the haem biosynthetic pathway, mitochondrial aconitase, an FeS enzyme which converts citrate to isocitrate in the Krebs cycle, and two proteins reputed to contribute to the pathology of Alzheimer’s disease and Parkinson’s disease, respectively, namely, amyloid precursor protein(APP) and α-synuclein. ROS will decrease the IRE-binding activity of IRP1 and IRP2, thereby activating storage and export of iron from cell. NO will increase binding activity of IRP1 but decrease that of IRP2 [7].

### 1.3. The Hepcidin-Ferroportin Regulatory System

The hepcidin-ferroportin regulatory system controls systemic iron homeostasis [10] by regulating iron release from macrophages, enterocytes and a number of other cells. The expression of hepcidin, a 25-residue peptide hormone produced by the liver (Figure 2), is regulated by many factors, including body iron stores, inflammatory factors (notably IL6) and requirements of iron for erythropoiesis, as well as hormones and growth factors. Replete iron stores increase the amount of the cytokines BMP2 and BMP6 (bone morphogenetic proteins 2 and 6) produced by liver sinusoidal epithelial cells, which initiate a regulatory cascade, resulting in hepcidin transcription. Hepcidin induces the internalisation and degradation of ferroportin, thus reducing the release of iron from duodenal enterocytes and macrophages of the reticuloendothelial system. In inflammation-related functional iron deficiency, hepcidin expression is increased via the IL-6, JAK2-STAT3 signalling pathway. In contrast, as a consequence of iron deficiency anaemia, the kidney produces erythropoietin, which increases erythropoiesis. When erythropoiesis is ineffective, hepcidin production in the liver is inhibited by erythroferrone (ERFE), which is produced in erythroblasts under the influence of EPO [11]. ERFE inhibits the BMP/SMAD signalling pathway by binding and inactivating BMP2 and BMP6 [12].

### 1.4. Brain Iron Metabolism

With regard to brain iron metabolism and homeostasis, we first consider how iron crosses the blood–brain barrier [14]. The brain itself is hidden behind a poorly permeable vascular barrier consisting of brain capillary endothelial cells (BCEC) of the cerebral blood vessels, and their associated pericytes, in contact with the end-feet of perivascular astrocytes and microglia which serve the nutritional requirements of a small number of local client neurons (typically less than 8), constituting the neurovascular unit. As with most human cells, it is likely that the major source of brain iron is Fe_2_Tf, which binds to TfR1 on the luminal face of the brain. The classic transferrin-to-cell cycle then transfers Fe(II) into the cytoplasm where it joins the intracellular LIP, while the ApoTf-TfR complex is transported back to the luminal membrane, where it is released. Iron can then be exported into the extracellular fluid of the brain by the Fe(II) exporter Fpn1, where it may undergo one of several fates. It could be taken up by the perivascular end feet of astrocytes, which ensheath the abluminal membrane of the BBB, in the form of low molecular weight complexes of iron with citrate, ATP or ascorbate, which astrocytes themselves secrete [15,16]. It could be loaded onto ApoTf, secreted principally by the choroid plexus, after oxidation to Fe(III) by ceruloplasmin, present in the CSF, or hephaestin, present along with ceruloplasmin in BCEC cells [17] and converted to Fe_2_Tf, which could deliver iron to those brain cells which express the TfR, such as neurons and microglia.

The ways in which both cellular and systemic iron homeostasis are regulated in the rest of the body have been discussed above, and it is interesting to note that, despite the overall homogeneity of these regulatory mechanisms, there are, nonetheless, cell-type specific regulatory mechanisms which are able to control or modify both local and systemic iron balance [18].

## 2. Neuroinflammation

Neuroinflammation in the central nervous system is primarily mediated by specific cell types, microglia, astrocytes, endothelial cells and pericytes. Peripheral inflammatory cells, such as macrophages, may also contribute to the inflammation, particularly if the brain–blood barrier integrity is compromised [19]. Neuroinflammation is co-morbid with brain iron accumulation, which is evident in many neurodegenerative diseases, e.g., Parkinson’s disease, PD; Alzheimer’s disease, AD; Huntington’s disease, HD; Friedreich’s ataxia, FA; and multiple sclerosis, MS. The aetiology of the iron loading remains unclear, although it is known that when present in glial cells and neurons, it will exacerbate the inflammatory process. The ability of iron to induce oxidative stress is a key pathological feature of neurodegeneration, whereby highly reactive radicals, such as hydroxyl radical are generated via the Fenton reaction causing damage to DNA, proteins, lipids, leading to cell death [20]. The term ferroptosis has been proposed to describe a type of programmed cell death dependent on iron and characterised by the accumulation of lipid peroxides which has been associated with the pathophysiology of neurodegenerative diseases.

### 2.1. Microglia

Brain-resident microglia originate from yolk sac macrophages during embryogenesis and are maintained and renewed from the population within the brain. Microglia show wide diversity and can exist in a variety of morphological forms; ramified, reactive, active or amoeboid, which will depend on their location in the brain as well as any activating stimuli, Figure 3. Ramified microglia are resting microglia, with small cell bodies and long processes which enable the microglia to monitor the environment via a range of receptors which include CX3C-chemokine ligand 1 (CX3CL1) CD47, CD200 and CD22. The interaction of CD200 with its receptor protein CD200R present on neurons leads to attenuation of a variety of immune responses [21]. Microglia show a diversity of functions ranging from promoting regrowth and remapping of damaged neurons through synaptic pruning, providing support for neurons and making direct contact with neuronal synapses, to remove neurotransmitters, which is achieved via a range of membrane receptors. Microglia continuously monitor the surrounding area and have a range of receptors which are sensitive to any changes in the extracellular microenvironment. Microglia are able to activate a variety of signalling pathways, MAPKs, DAMP, RAGE, TLR via the transcription factor, NFkappaB, to generate reactive nitrogen species (NO), and NADPH oxidase, to generate reactive oxygen species (ROS) Figure 4. In normal circumstances such activation would be transient and be resolved as basal physiology is restored, but during neuroinflammation, as seen in neurodegenerative diseases, the microglia remain chronically activated. Activation of microglia via TLRs and interferon signalling pathways will induce the release of pro-inflammatory cytokines and chemokines such as tumour necrosis factor-alpha (TNF-α), cytokines including interleukin-6 (IL-6), interleukin 1-beta (IL-1β), interleukin-12 (IL-12) and C-C Motif Chemokine Ligand 2 (CCL2), as well as NO via iNOS activation. Recovering microglia will secrete anti-inflammatory cytokines, such as interleukin-10 (IL-10) and transforming growth factor-beta (TGF-β). In addition, the cells can secrete a variety of factors including insulin-like growth factor I (IGF-I), fibroblast growth factor (FGF) and neurotrophic factors including nerve growth factor (NGF) and brain-derived neurotrophic factor (BDNF), in the effort to resolve inflammation and promote synaptic plasticity. Activated microglia exhibit a directional migration towards the site of pathology, by using a chemo-attractant gradient as a directional cue which will recruit additional microglia to the area of damage [22]. In early studies, these two states of microglia were described as M1, the pro-inflammatory state, and M2, the -anti-inflammatory state. However, this clearly was an oversimplification of such processes in that there is constant gradation of microglia phenotypes.

Iron uptake by microglia is by the (Tf-TfR) pathway, and by DMT1, export via FPN1 [23] and storage within the cytosolic iron protein, ferritin. The expression of these proteins is translationally regulated by the iron responsive element/iron regulatory protein (IRE/IRP) system, which in normal circumstances is activated when the cell has a low iron level, thereby increasing DMT1 and TfR1 levels and decreasing FPN1 and ferritin expression. Iron loading of these cells will lead to deregulation of brain iron homeostasis.

### 2.2. Astrocytes

Astrocytes are the most populous glial cell and are critical for brain function, being essential for maintaining neuronal health. They provide structural and metabolic support and regulate synaptic transmission, water transport and blood flow in the brain. Astrocytes produce various neurotropic molecules, including glial-derived neurotropic factor and glutathione, the latter being transferred to neurons which are unable to synthesise this tripeptide. Astrocytes play an important role at the BBB, having specialised end feet that extend from the astrocyte cell body and attach to the basement membrane that surrounds the endothelial cells and periocytes on the brain vasculature. Astrocytes control blood–brain permeability and maintain extracellular homeostasis. Astrocytes also play an important role in the glymphatic system, forming perivascular tunnels, to promote efficient elimination of soluble proteins and metabolites from the central nervous system. Abnormal proteins, such as α–synuclein and β-amyloid, are drained out of the brain by this process, during sleep, thereby emphasising the importance of sleep [24]. Activated astrocytes are characterised by a different molecular pattern, morphology, becoming reactive, scar-forming astrocytes and hypertrophic astrocytes [25], which alter many of their homeostatic functions, e.g., Ca^2+^ signalling, excitatory neurotransmitter uptake. Reactive astrocytes will secrete a wide number of cytokines and ROS. There is cross-talk between astrocytes and microglia, via a number of secreted mediators, e.g., growth factors, neurotransmitters and gliotransmitters, cytokines and chemokines.

Astrocytes play an important role in brain iron homeostasis as many are positioned close to the blood–brain barrier and take up iron from the circulation and distribute iron to other cells in the CNS as described earlier. Iron influx mechanisms include DMT1 and the Tf-TfR mediated process. The glycosylphosphatidylinositol-anchored form of caeruloplasmin is highly expressed by astrocytes and is physically associated with ferroportin. It has been proposed that hepcidin secreted by astrocytes could regulate FPN1 on brain microvascular endothelial cells (BMVECs), inducing the internalisation and degradation of FPN1, and, thus, control iron transport across the BBB. It remains unclear whether astrocytes are able to accumulate and store iron.

IRPs 1 and 2, DMT1+ and IRE are expressed by astrocytes. Increases in IRP regulatory protein expression could lead to elevation of the +IRE form of DMT1. The pro-inflammatory cytokine IL-6 is one of the strongest positive regulators of hepcidin mRNA expression via the activation of the JAK/STAT3 pathway. Early studies indicated that the release of IL-6 from activated microglia initiated an intercellular l cascade where this cytokine stimulates astrocytes to release hepcidin which, in turn, signals to neurons, via the hepcidin-FPN1 axis, to prevent their iron release. Thus, iron storage within brain cells would be enhanced by internalisation of the ferroportin-hepcidin complex.

### 2.3. Oligodendrocytes

Oligodendrocytes are the myelinating cells of the CNS and their myelin sheath wraps around axons to allow fast saltatory conduction of action potentials. A wide range of cytokines and chemokines are expressed and upregulated in response to inflammation. Indeed, oligodendrocytes express several receptors for immune molecules which are able to respond to inflammation and react appropriately. Since large amounts of iron are required for axon myelination, oligodendrocytes are rich in iron and take up iron, via a ferritin receptor, known as Tim-1 in human and Tim-2 in rodents, DMT1 or other non-vesicle import mechanisms. Oligodendrocytes might also extract iron from adjacent blood vessels. Iron is mostly found within ferritin and transferrin in oligodendrocytes.

### 2.4. Neurons

Neurons, the main component of nervous tissue in the brain, are electrically excitable cells that communicates with other neurons via synapses. A typical neuron consists of a cell body, soma, dendrites and a single axon. One neuron affects other neurons by releasing a neurotransmitter that binds to chemical receptors, the two most common neurotransmitters in the brain being glutamate and GABA, (>90%). Other types of neurons include cholinergic, adrenergic, dopaminergic, serotonergic and purinergic. Neurons show a high density of FPN compared to other brain cells which might explain why FPN downregulation causes more marked iron loading in neurons [26]. Furthermore, cellular iron overload seems to be more detrimental to neurons due to their lower iron buffering capacity [27]. Iron released into brain parenchyma will enter the neuronal cells via Tf-TfR receptor mediated process, the transient receptor potential canonical (TRPC) channel, DMT1 and Zip8 and Steap2.

A graphical overview of the iron proteins involved in glia and neuronal iron metabolism is presented in our previous publication [15].

## 3. Neuroinflammation Mediated Neurodegeneration in the Brain

The stimulus for the initiation of the inflammatory process in the brain in neurodegenerative diseases is unclear, but once glial cells such as microglia and astrocytes are activated, the response is chronic and sustained. Iron dyshomeostasis is also apparent, playing a pivotal role in sustaining the neuroinflammtory phenotype. There are many factors which could be contributing to the sustained inflammatory process.

### 3.1. Aging

With aging, the inflammasome (innate immune system receptors and sensors that regulate caspase-1 activation), becomes activated, thereby modulating caspases, which will lead to an inflammatory response with the release of pro-inflammatory mediators, complement components and adhesion molecules [28]. Known as inflammaging, this could affect protein folding. Various molecules involved in iron homeostasis are altered with aging: HO-1 an enzyme which degrades haem, releasing several molecules including carbon monoxide and ferrous iron [29], while abluminal hepcidin is reputed to be upregulated in aged rats. Such changes would increase iron accumulation in brain. The ability of microglia and astrocytes to maintain homeostatic equilibrium for neurons is reduced with aging. Interferon-induced transmembrane protein 3, IFITM3, (a protein that is released in the immune response to pathogens), is elevated which will increase the activity of gamma secretase, an important enzyme which cleaves APP to form Aβ, the precursor of the amyloid plaque. Proteins such as major histocompatibility complex class II (MHCII), integrins and toll like receptors on microglia are upregulated allowing for a stronger pro-inflammatory response. There is a loss of normal astrocyte function, i.e., the promotion of neuron survival and synapse formation. Normal aging is accompanied by progressive iron accumulation in the brain, primarily in the substantia nigra, putamen, globus pallidus, caudate nucleus, and cortices, as ferritin and neuromelanin. Interestingly, in a study where local systemic inflammation was induced by an acute lipopolysaccharide, LPS, injection into the striatum of aged rats, the iron content and ferritin expression increased in the SN, microglia were activated and lipid oxidative stress was evident [30].

### 3.2. Protein Misfolding

Aggregation and deposition of misfolded proteins occur in many neurodegenerative diseases. Correct folding of proteins requires the protein to assume its unique tertiary structure from a constellation of possible, but incorrect, conformations. Elaborate systems have evolved to protect cells from the deleterious effects of misfolded proteins, i.e., molecular chaperones to promote correct folding, and a ubiquitin-proteasome/autophagy systems to remove abnormally folded proteins. When this system is overwhelmed, misfolded proteins will accumulate, e.g., tau and Aβ protein in AD, and alpha-synuclein in PD. Neurons are particularly sensitive to the toxic effects of such inclusion bodies, recognising that the altered proteins (oligomers) are toxic, and they export them out of the cell. This can lead to microglial activation and pathological spread since adjacent neurons take up the altered protein and this triggers a prion like reaction within the healthy neuron. Such toxicity will induce inflammation which will gradually spread across different brain regions. The explanation as to why proteins take this route remains unclear, whether this is a direct or a response to an external stimulus remains unknown.

### 3.3. Mitochondria

Mitochondria play an important role in energy production via oxidative phosphorylation, as well as being involved in the synthesis of haem Fe-S proteins. During ageing, the autophagy-lysosomal system for removing defective mitochondria becomes less effective resulting in greater ROS production, mtROS and lower ATP production. Mitochondria are also very sensitive to misfolded alpha-synuclein which enters the mitochondria and inhibits Complex 1. This results in an energetic crisis with increased levels of products of oxidative damage, such as 8-OH-dG and mutation load in mtDN. Such ROS could be involved in triggering inflammation [31] and altering iron homeostasis.

### 3.4. Gut

Cross-talk between the gut microbiota and the CNS, has recently been a major topic of interest as this may play a major initiating role in the pathogenesis of many age related neurodegenerative diseases, such as PD, AD, ALS and HD [32]. The mechanism by which bacteria induce protein aggregation and contribute to the pathogenicity of protein misfolding remains unclear, although changes of the bacteria flora to a more pro-inflammatory phenotype may lead to local inflammation within the gut wall. Experiments in mice have identified a particular strain of *E. coli*, which synthesises a protein named curli, which prompts proteins to misfold, and can then be transmitted via the vagus nerve to the brain. Experimental models of neurodegeneration, e.g., [33] have shown that systemic microbial infections or bacterial sepsis can lead to enhanced amyloid load, neuroinflammation and cognitive impairment in transgenic models of AD.

### 3.5. Peripheral Inflammatory Markers

Systemic proinflammatory cytokines, and immune and inflammatory mediators can promote a proinflammatory environment in the CNS by crossing the BBB, by signalling via endothelial cells or circumventricular organs. Animal studies have shown that peripheral inflammation augments neuroinflammatory pathways by activating glial cells, as well as increasing BBB permeability [34]. The vagus nerve may also signal inflammatory proteins via direct afferent connections to the brain stem. The absence of this inflammatory reflux will result in excessive innate immune responses and the release of proinflammatory cytokines [35].

### 3.6. Blood–Brain Barrier

The role played by the blood–brain barrier, the brain microvascular endothelial cells, in response to the systemic inflammation, and their ability to transfer iron remains unclear. Functional changes in iron uptake and efflux proteins may occur. Since inflammation would increase circulating hepcidin, decreased efflux of iron via ferroportin could occur. Iron uptake into microvascular endothelial cells, may be regulated via ZIP proteins, ZIP14 and ZIP18, inflammation, specifically IL6, which can upregulate Zip14 expression [36]. It remains unclear how inflammation may affect BBB permeability via tight junctions and adherens junction, CRP may play a role in barrier breakdown [37,38]. The function of astrocytes, present on the abluminal side of the BBB, will be altered by inflammation, and will release glutamate, an excitotoxic neurotransmitter that exacerbates the inflammation.

## 4. Neurodegenerative Diseases with Special Emphasis on PD and AD

In the two neurodegenerative diseases discussed in this paper, PD and AD, there is a sustained neuroinflammatory process which alters iron homeostasis. Interestingly the changes in iron deposition in the two diseases differ widely; in PD, there is a dramatic increase in iron, and in AD, only a marginal increase in iron is apparent. The question that remains is what the stimulus for the increased iron deposition is. Since there is a close relationship between iron homeostasis and inflammation, one might presume that inflammatory changes, with the release of ROS and NO as well as specific cytokines, will affect the iron proteins involved in iron homeostasis. The upregulation of hepcidin by inflammatory stress is a major critical event triggering iron incorporation into cells via down-regulation of ferroportin [39]. Increases in IL-6 in the circulation or brain would promote cellular iron accumulation as hepcidin will bind to ferroportin. Furthermore, TNF–α, a pro-inflammatory cytokine will also induce iron accumulation in cells independent of the induction of hepcidin, possibly via promotion of DMT1 [39]. Therefore, in the following examples of neurodegenerative diseases, namely, Parkinson’s Disease and Alzheimer’s Disease, we will evaluate whether there is a systemic inflammatory response which could be responsible for the neuroinflammation and which in turn would alter iron homeostasis.

### 4.1. Parkinson Disease

PD is the most common form of motor system degeneration. PD affects approximately 1% of the aged population over the age of 60 and about 4% above 85 years old. Symptoms of the disease include bradykinesia (slowness of movement), rigidity, resting tremor and postural instability. PD is primarily considered a motor disorder but there are also a whole host of non-motor symptoms, such as sleep disturbance, anxiety, depression, cognitive decline, etc., many of which occur before the onset of motor deficits. There is misfolding of the protein α-synuclein which results in the deposition of insoluble protein inclusions in neurons, Lewy bodies. The toxic oligomeric forms of a-synuclein are released from the dying neurons, and will be phagocytosed by surrounding microglia. These will become activated and drive the inflammatory process. In vitro, it has been shown that microglia loaded with a-synuclein can transfer the protein to adjacent naïve microglia [40] possibly accentuating the inflammation.

#### 4.1.1. Peripheral Circulation

Epidemiological studies reported a correlation between systemic inflammatory events, chronic neuroinflammation and the aetiology and progressive nature of PD. Increased serum levels of pro-inflammatory cytokines, monocytes, neutrophils leucocytes and C-reactive protein, CRP [41], have been assayed in PD patients, indicating a heightened inflammatory response. Hepcidin levels are increased in the serum and positively correlate with serum ferritin, an acute phase reactant, in PD patients of less than 5 years disease duration [42]. Lymphocyte infiltration has been observed in brains of PD patients, which could promote/reinforce microglial activation [43]. It was of interest that a H1N1 influenza-A pandemic was coupled with a dramatic increase in post-encephalitic Parkinsonism (PEP) (also referred to as “sleeping sickness” or von Economo encephalitis) at the end of the First World War [44]. It was shown that that H1N1 virus preferentially targets the SNpc, the primary site of pathology in PD [45].

#### 4.1.2. Mitochondria Function

A multitude of mitochondrial defects are caused by toxic α-synuclein oligomeric forms which include decreased mitochondrial membrane potential and energy production, disruption of mitochondrial-ER Ca^2+^ homeostasis, inhibition of mitochondrial dynamics and induced mitochondrial pro-apoptotic protein cytochrome c release [46]. Increased cellular ROS will occur and there will be mitochondrial respiratory Complex I dysfunction.

#### 4.1.3. Blood–Brain Barrier, BBB

BBB is thought to be intact in PD, although recent studies have reported some disruption [47]. The severity of the disease may be an important factor.

#### 4.1.4. Neuroinflammation

Since McGeer et al. [48] reported the presence of activated microglia and T-lymphocytes in the SN of post mortem brain, this has been confirmed in many studies. With progression of the disease, microglia activation is present in many other regions including putamen, hippocampus and cortex [20].

Intense microgliosis around extraneuronal neuromelanin (released by dying neurons) has been identified in the SN of PD patients. In vitro and in vivo studies have confirmed the ability of neuromelanin to activate microglia and induce death of dopaminergic neurons [49]. The release of toxic misfolded α-synuclein from dying neurons can potently activate microglia.

Evidence is emerging to suggest that disruption of astrocyte biology is involved in dopaminergic neuron degeneration in PD. Gene mutations implicated in PD, e.g., 3A2, PINK1, DJ-1, α-synuclein, iPLA_2_ and ATP1rkin, may affect astrocyte function, such as inflammatory responses, glutamate transport, and neurotropic capacity [50].

#### 4.1.5. Iron in PD and Changes in Iron Proteins

Semi-quantitative histochemical methods showed that iron deposits were present in the neurons and glia of the substantia nigra, putamen, and globus pallidus, with an increase of ferritin-loaded microglia cells in the substantia nigra [51]. The increased iron content found in the SN is associated with microglia and dopaminergic neurons. Indeed, in our recent study, a significant correlation was found between iron accumulation and microglial intensity in the SN of post mortems of PD brains [51], suggesting a link between the extent of iron loading and neuroinflammation. Excesses of iron are engulfed by neuromelanin within neurons, which will be released into the extraneuronal space when the neuron dies, thereby activating microglia.

Quantitative analysis of PD brains by micro particle induced Xray emission showed that the iron concentration was increased in most cell types in the SN, except for astrocytes and ferritin-positive oligodendrocytes. The highest cellular iron levels in neurons were located in the cytoplasm, which might increase the source of non-chelated Fe^3+^. [52].

Various alterations in specific iron proteins which have been reported, such as increased expression of DMT1 in dopamine neurons [53], changes in ferritin production due to sustained IRP1 activity, increased DMT1 activity or decreased ferroxidase activity of ceruloplasmin [54], could be attributable to inflammatory induced alteration in iron homeostasis.

### 4.2. Alzheimers Disease

Alzheimer’s disease (AD) is the most common form of dementia, and accounts for approximately 80% of all dementia cases. It is a disease that occurs primarily in people >70 years old and is characterised by progressive memory loss and reduction in higher cognitive functions, caused by the loss of synapses, particularly in the hippocampus and entorhinal cortex. The AD brain shows moderate cortical atrophy, the frontal and temporal cortices often have enlarged sulcal spaces with atrophy in posterior cortical areas, the precuneus and posterior cingulate gyrus. This often results in enlargement of frontal and temporal horns of the lateral ventricles.

The presence of beta-amyloid (Aβ) plaques, in the extracellular compartments and neurofibrillary tangles, NFT, intracellularly, are the characteristic pathology of AD. The Aβ plaques are formed as a result of improper cleavage of APP, which results in Aβ monomers that will aggregate to form oligomeric Aβ. These will aggregate into Aβ fibrils and plaques. NFTs occur due to the phosphorylation of tau at multiple sites which results in the disruption of several cellular processes which range from protein trafficking to overall cellular morphology. These two pathological features are often present for many years before the diagnosis of AD. A recent study by Pascoal et al. [55] suggested that an interaction between neuroinflammation and amyloid pathology induced tau propagation, microglia activation and inflammation which resulted in widespread brain damage.

#### 4.2.1. Peripheral Circulation

Elevated levels of pro- and anti-inflammatory mediators, e.g., IL-1β, IL-2, IL-6, IL-8 IL-10, IL-12, IL-18, interferon-γ, TNF-α, MCP1 and transforming growth factor-β, have been assayed in the blood and CSF of AD patients [56]. Serum TNF-α levels are elevated in AD, which is associated with faster progression of the disease. Serum iron was significantly lower in AD by comparison to controls [57], possibly indicating the anaemia of chronic disease. These inflammatory mediators will induce reactive pro-inflammatory microglia and astrocytic phenotypes in the brain, which will promote Aβ oligomerisation, and complement activation. Such changes may initiate or exacerbate neurodegenerative processes leading to cognitive decline and dementia.

#### 4.2.2. Neuroinflammation in AD

PET scanning with [^11^C]DPA713 tracer identified microgliosis in many brain regions of AD brains [58]. The use of first generation tracers, such as 11 C(R0-PK11195, identified the 18 kDa translocator protein (TSPO) (known as the peripheral benzodiazepine receptor which is predominately localised to the outer mitochondrial membrane) in activated microglia and is non-specific [59]. Therefore, there is an urgent need for more specific microglia marker tracers. Microglia will be activated after phagocytising Aβ with the release of a number of pro-inflammatory markers [60]. Aβ will bind to various receptors on microglia, CD36, TLR4 and TLR6, which will be further activated with the release of a number of pro-inflammatory cytokines and chemokines. IL-1β may be the master regulator of the brain inflammatory cascade due to its integral role in regulating the expression of other proinflammatory cytokines, e.g., TNF-α and IL-6. Disruptions to IL-1β may delay the onset of neuroinflammation and neurodegeneration [61].

Hypertrophic reactive astrocytes accumulate around senile plaques, (identified by increased glial fibrillary acidic protein staining) and may show functional impairment [62]. Astrocytes will release cytokines, nitric oxide and other proinflammatory markers to exacerbate the neuroinflammation. Although astrocytes can upregulate a range of Aβ degrading proteases, this may be impaired in these reactive astrocytes such that their ability to proteolytically clear Aβ is reduced (reviewed in [63]).

#### 4.2.3. Mitochondria

The amyloid cascade hypothesis indicated that Aβ was responsible for triggering downstream tau pathology. Tau is present predominantly in axons and plays a role in axonal transport, such that accumulation of pathological tau might affect mitochondrial transport and cause mitochondrial dysfunction. However, it is unclear whether Aβ directly affects mitochondrial bioenergetics and produces mitochondrial morphological change [64].

#### 4.2.4. Blood–Brain Barrier

Leakage of plasma proteins that associate with the senile plaques has been identified in post mortem AD brains, indicating that there is increased permeability of the BBB [65]. P-glycoprotein plays an important role in the clearance of cerebral β-amyloid (Aβ) across the BBB. Using positron emission tomography, significantly lower P-glycoprotein was identified in the parietotemporal, frontal, posterior cingulate cortices and hippocampus of mild AD subjects.

#### 4.2.5. Iron Loading and Iron Proteins in AD

Increases of iron in brain regions, including the substantia nigra, globus pallidum, hippocampus, putamen and caudate nucleus have been reported and are related to brain cognitive and memory functions [57]. Iron is associated with senile plaques and neurofibrillary tangles or with ferritin in the surrounding glia cells (Reviewed by [66]). Such iron deposition could be related to many factors, e.g., increased permeability of the blood–brain barrier, dilation of blood vessels, redistribution of iron and iron homeostasis changes or neuroinflammation [15]. Iron can induce the production and accumulation of amyloid-β since iron will influence the proteolytic activation of the inactive forms of α-secretase and β-secretase which is modulated by furin [67]. Excess iron decreases furin protein concentration, which will favour β-secretase activity and stimulate the amyloidogenic pathway to produce Aβ [15,67]. Iron may modulate APP processing since there is a putative IRP element in APP. The 5′-UTR of APP mRNA possess a functional IRE stem loop with homology to IREs in ferritin and transferrin receptor mRNA located immediately upstream of an IL-1 responsive box domain [68]. Increases in IL-1 production, increases IRP binding to the APP 5′UTR thereby decreasing APP production [68]. In early studies of three cerebral cortical regions of post mortem AD material, transferrin was consistently decreased in the white matter [69], while no significant alterations in iron and ferritin changes were detected. The decrease in transferrin levels could indicate a decreased mobility and subsequent utilisation of iron in the brain and play a significant role in neuronal degeneration and increased peroxidative damage. However, as yet, no correlation between iron accumulation and neuroinflammation has been reported. Interestingly, despite the extensive neuroinflammation in AD brains, the increase in iron is not as dramatic as that observed in PD brains. Further studies to ascertain why iron levels are not dramatically increased are clearly warranted.

## 5. Therapeutic Approaches

### 5.1. Non-Steroidal Inflammatory Drugs, NSAI

Beginning in the 1990s, several large epidemiological and observational studies were published indicating that anti-inflammatory treatments used in diseases, such as rheumatoid arthritis, showed protective qualities against developing AD or PD. Indeed, it was demonstrated that there was a 50% reduction in the risk for developing AD in patients who are long-term non-steroidal anti-inflammatory drug (NSAID) users. These studies led to studies utilizing animal transgenic AD models demonstrating that NSAIDs can reduce AD pathology. Currently, there are many epidemiological studies that indicate that anti-inflammatory drugs may protect against AD (reviewed by [70]). Although other human trials of NSAIDS showed variable outcomes with no convincing evidence of benefit, it has been suggested that a better refinement of treatment strategy is required, and particular attention should be directed to the selection of patients and intervention stage timing.

### 5.2. N-Acetyl Cysteine

Studies have shown that NAC amplifies glutathione levels, increased levels of anti-oxidants and decreased expression of IL-6, TNFα and IL1β in aged rats [71]. One small study, where 42 PD patients randomly received either weekly intravenous infusions of NAC (50 mg/kg) plus oral doses (500 mg twice per day) for 3 months or standard of care only, showed that the dopamine transporter binding significantly increased in the NAC group in the caudate and putamen (mean increase from 3.4% to 8.3%) compared with controls. This was associated with significantly improved PD symptoms [72].

### 5.3. Iron Chelation

Iron chelation therapy was first introduced for the treatment of thalassemia major in 1963, with the hexadentate chelator, deferrioxamine (DFO). Successful therapy of this congenital disease requires both blood transfusion to combat the anaemia and chelation therapy to remove the accumulation of body iron which is the consequence of the transfusion regime. However, the very short plasma half-life and poor oral activity of DFO led to the development of two new orally active iron chelators for the treatment of transfusional iron overload, the bidentate deferiprone (DFP) and the tridentate deferasirox (DFX). All three of these chelators, which are currently approved for clinical use, were shown to be neuroprotective in an animal model of PD [73]. While the potential of iron chelation therapy to protect from the consequences of iron overload and toxicity in neurodegeneratative diseases such as PD and ALS (amyotrophic lateral sclerosis) has been based largely on studies on cellular and animal models (reviewed in [74]), to date, only DFP has been used in clinical trials in PD and AD. Two clinical trials were initiated in 2011 and 2013 to examine whether a 6–9-month treatment with DFP could reduce the iron load in the SN and affect the severity and progression of the disease in Parkinson’s patients [42,75]. MRI techniques were used to evaluate iron content in the SN, UDPRS scores to follow the evolution of the disease and serum ferritin as a marker of iron stores and inflammation, and both studies established the efficacy of chelation therapy. However, in both studies, it was necessary to monitor the participants on a weekly basis for adverse effects (agranulocytosis), which was observed in a small number of participants warranting drug withdrawal; neutrophil numbers returned quickly to within normal values after deferiprone withdrawal. It should be noted that in the Imperial London study [42], after withdrawal of DFP at the end of the Clinical Trial, iron levels returned to pre-treatment values relatively rapidly. It was also noted that in patients who responded to iron chelation, serum levels of ferritin also decreased possibly indicating a reduction in the acute phase response [41].

In the FAIR PARK II (EudraCT number 2015-003679-31), randomised, double-blind trial, participants were randomly assigned in a 1:1 ratio to receive either oral deferiprone or matched placebo for 36 weeks followed by a 4-week post-treatment monitoring period. The primary criterion was a change to the patient’s total score on the Movement Disorder Society (MDSUPDRS) through 36 weeks. Secondary outcomes were the motor and non-motor handicap, the iron content and volume measured by Magnetic Resonance Imaging and the dopamine transporter (DaT) density.

The first study results became available recently (Fairpark II, January 2022, [76]). A total of 411 persons were screened for eligibility, and 372 were randomly assigned to receive deferiprone (186 participants) or placebo (186 participants). In an intention-to-treat analysis, the mean change in the total MDS-UPDRS score was 16.7 points with the active drug and 6.3 points with placebo (difference, 10.4 points; 95% confidence interval, 6.8 to 14.1; P). The iron content in the nigro-striatal pathway was significantly reduced under deferiprone. A significant change in the volume of striatum was observed with a smaller volume in the placebo group. No change of the DaT density was observed. Serious adverse events with deferiprone were agranulocytosis, as observed on the two earlier clinical trials.

Deferiprone was found to worsen the handicap of persons without dopaminergic treatment as compared with placebo. The authors conclude that longer trials in persons under dopaminergic treatments are necessary to evaluate whether deferiprone could reduce disease progression.

A multi-centre study is in progress in Australia (Deferiprone to Delay Dementia (The 3D Study)) in January 2018 to compare the efficacy of Deferiprone (15 mg/kg) administered orally twice a day with a matching placebo in subjects with pAD (MCI with brain amyloid pathology) or mAD at 12 months relative to baseline. The study has recruited 171 patients and is set to reach completion by September 2023.

## 6. Conclusions

The hypothesis advanced at the start of this review was that the iron loading in specific brain regions in PD and AD may be a consequence of an initial neuro-inflammatory event. Thus, the release of a number of cytokines and reactive oxygen and nitrogen species from activated glial cells are able to alter the activity of many iron proteins involved in maintaining iron homeostasis. The observation that PD patients improved their brain iron status after chelation therapy, but that it was rapidly reversed upon cessation of chelator treatment, would be entirely consistent with our hypothesis. Therefore, as we have previously suggested, a judicious combination of anti-inflammatory and chelation therapy might be the most appropriate therapeutic approach.

## Figures and Tables

**Figure 1 ijms-23-07267-f001:**
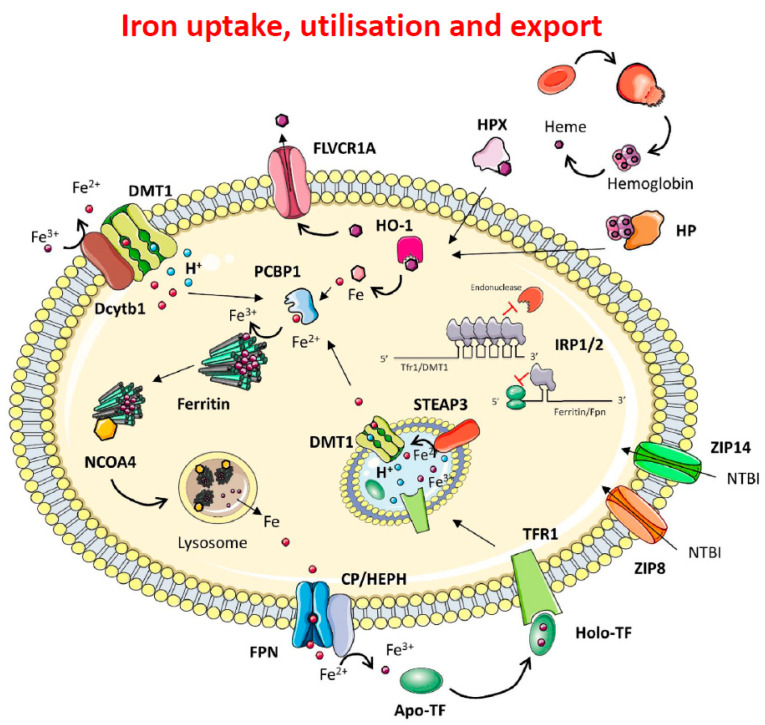
Mechanisms of iron import, handling and export. Dietary ferric iron is reduced to ferrous iron by DCYTB and imported via DMT1. Once in the cytosol, iron is chaperoned by PCBP1 and delivered to Ferritin for storage. In iron depletion conditions, NCOA4 mediates lysosomal ferritinophagy and cellular iron release via FPN. Iron is oxidised to ferric iron via Ceruloplasmin or Hephaestin and loaded onto transferrin to be delivered via the bloodstream to all cell types. In the circulation, diferric Tf (Fe_2_Tf binds to the transferrin receptor (TfR1), and once bound to TfR1, the TfR1/Fe_2_-Tf complex is internalised into an endosome which is acidified by a proton pump to a pH of 5.6. Under these conditions Fe^3+^ is released from the complex and is reduced to Fe^2+^ by the reductase Steap3. The Fe^2+^ is then transported into the cytoplasm by Divalent Metal Transporter 1 (DMT1), and the ApoTf/TfR1 complex is returned to the cell surface where it is released back into the circulation for reutilisation. As we will see later, there is a second transferrin receptor, TfR2, which is involved in systemic iron homeostasis. Cellular iron homeostasis is orchestrated via the IRE/IRP system that regulates the expression of TFR1 and DMT1, and of both chains of ferritin and the iron exporter Fpn1. Non transferrin bound iron (NTBI) is imported via the Zinc–Iron-Proteins ZIP8 and ZIP14. In haemolytic conditions, haemoglobin and haem released into the circulation are respectively bound to haptoglobin (HP) and haemopexin (HPX) to be internalised by the cell. HO-1 cleaves haem to form biliverdin and releases ferrous iron. Newly synthetised haem is exported via FLVCR [6].

**Figure 2 ijms-23-07267-f002:**
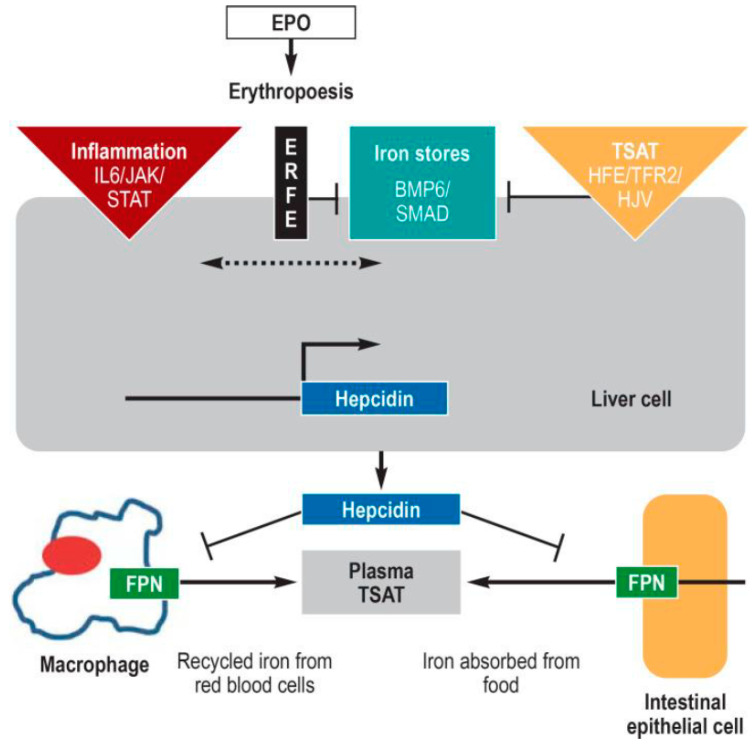
The 25-residue peptide hormone, hepcidin, produced by the liver, is the central regulator of systemic iron metabolism. When hepcidin is released into the circulation, it binds to the iron exporter, ferroportin, resulting in internalisation and degradation of ferroportin, thereby preventing the release of iron from duodenal enterocytes and macrophages of the reticuloendothelial system. Hepcidin expression is controlled by several factors: (i) the amount of available iron, represented by body iron stores (centre) and transferrin saturation [TSAT] (right); (ii) inflammation, essentially via the proinflammatory cytokine, Interleukin 6 [IL6] left); and (iii) the iron requirement for erythropoiesis (centre). Replete iron stores increase the amount of the cytokines BMP2 and 6 (bone morphogenic protein 2 and 6) which, together with the cofactor hemojuvelin (HJV), activates the BMP/SMAD signaling pathway, resulting in hepcidin induction. In inflammation-related functional iron deficiency, hepcidin expression is increased via the JAK/STAT signaling pathway. As a function of the functional iron deficiency, the kidney produces erythropoietin [EPO], which increases erythropoiesis. The erythroid precursor cells continue to produce erythroferron [ERFE], which inhibits the BMP/SMAP pathway, decreasing hepcidin expression. Abbreviations not defined above HFE, high iron; JAK Janus kinase; STAT, signal transducers and activators of transcription; SMAD, small-body-size mothers against decapentaplegic homolog 1; TFR2 transferrin receptor 2 [13].

**Figure 3 ijms-23-07267-f003:**
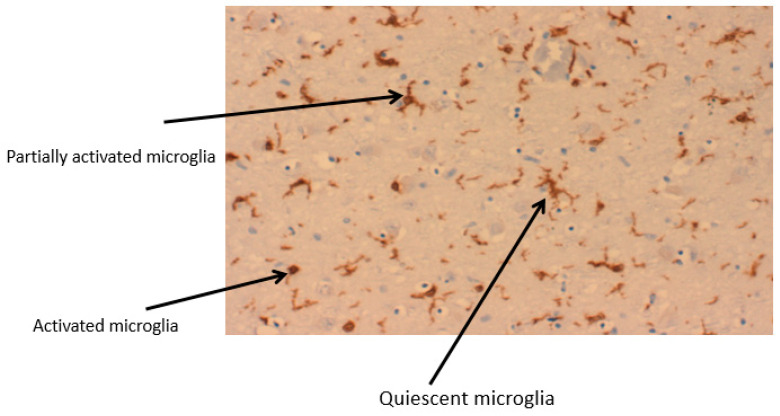
Stages of microglia activation from a ramified structure to an amoeboid form. A section of a Parkinson’s brain, of the region, substantia nigra showing 3 different stages of microglia activation. IbA staining.

**Figure 4 ijms-23-07267-f004:**
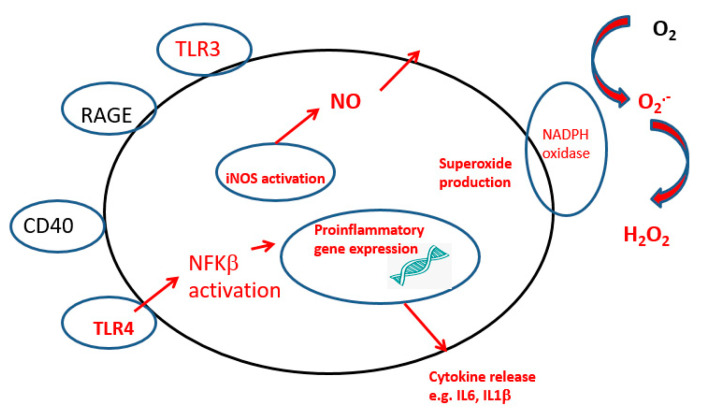
Activated microglia showing receptors involved in the instigation of the inflammatory process. DAMP, Damage-associated molecular pattern molecules TLR. Toll-like receptors; NADPH, nicotinamide adenine dinucleotide phosphate oxidase; RAGE, receptor of advanced glycation end-products; CD, cluster of differentiation. The major inflammatory pathway is marked in red.

## Data Availability

Not applicable.

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
