# Peer review of "Iron, Neuroinflammation and Neurodegeneration"

_ijms, 2022, doi:10.3390/ijms23137267_

Round 1

Reviewer 1 Report

The authors propose a careful and detailed review regarding iron metabolism, as well as neuroinflammation due in particular to the main pathologies affecting the CNS.

Most likely the action of iron is consequent, it does not cause inflammation, although it probably has the ability to accentuate it.

Perhaps this aspect could be better emphasized by proposing therapeutic alternatives that act directly on iron and not on inflammation.

In general, the parts concerning neuroinflammation and pathologies should be better focused, even reducing them, on the main theme, namely iron.

Please clarify the legend in fig 2. It is understandable what it wants to communicate, but it is difficult to read.

Author Response

Reviewer 1.

  1. We agree that iron would accentuate the generation of ROS and this stated in the text
  2. In the section on therapeutics there is a large section devoted to iron chelation and the results of the 2 clinical trials to date. A sentence has been inserted ‘. It is was also noted that in patients who responded to iron chelation, serum levels of ferritin also decreased possibly indicating a reduction in the acute phase response [41] to show the connection between iron chelation and inflammation
  3. We disagree with the reviewer’s statement that the section pertaining to neuroinflammation and pathologies should be reduced. It is of importance to emphasise that there is a relationship between iron, neuroinflammation and neurodegeneration in many of these diseases and that a joint therapeutic approach should be used, that is to reduce both inflammation and iron content in the brain

The legend of Figure 2 has been rewritten for clarity sake

Reviewer 2 Report

The Authors describe how the disruption of brain homeostasis, either directly through the creation of aberrant proteins or indirectly through peripheral inflammation, will trigger microglia to synthesize a range of pro-inflammatory chemicals, potentially leading to inflammation and cell death, causing alterations in the iron proteins necessary for regulating iron homeostasis thus resulting in increased iron deposits in brain cells. The production of reactive oxygen and nitrogen species, which are directly implicated in the inflammatory process, can have a major impact on iron metabolism via interactions with iron-regulatory proteins (IRPs). This supports the necessity of maintaining iron in a regulated form, as evidenced by the sophisticated mechanisms that have become more well known for controlling iron homeostasis. N-acetyl cysteine, nonsteroidal anti-inflammatory drugs, and iron chelation are among the therapeutic techniques used to reduce iron toxicity.

The topic is important. Overall, the manuscript is well written and of interest.

Minor point.

In my file, downloaded by the IJMS platform, figure 3 has a low resolution and is of poor quality.

Line 297, inflammasone, please, correct the term

Through the manuscript, Abeta is A, please, correct it

Author Response

Reviewer 2.

  1. Figure 3 is stated to have low resolution and quality. This has been rejigged
  2. Inflammasone has been corrected to inflammasome
  3. I do not know why many of the A beta has been altered. In the original submitted paper they were correct. These  has now been corrected

Reviewer 3 Report

This is a very comprehensive review highlighting the salient features of Iron metabolism and homeostasis by different brain cell populations. The review is very precise and discuss perturbation in iron homeostasis in the context of neuropathological disorders such as Alzheimers's disease and Parkinson disease. The different mechanisms of Iron induced cellular death has been clearly presented.

Comments:

1. I would suggest proof reading of the entire manuscript as there are a number of places where spacings and sentence integrity is inconsistent.

2. Graphical representation of the differences in iron metabolism in neurons, astrocytes, microglia, and oligodendrocytes.

Author Response

  1. We have reviewed the entire manuscript to correct spacing and sentence integrity
  2. We think that the second statement,’ Graphical representation of differences in iron metabolism in neurons, astrocytes, microglia and oligodentrocytes’ is suggesting that a diagram to show the differences in iron proteins in the different cell types should be shown. This we have shown in one of our previous papers. We have therefore added a line to direct the reader to our previous paper where a graphical representation is given.